# Comparing Different Methods for Pruning Pitaya (*Hylocereus undatus*)

Emilio Arredondo, Fernando M. Chiamolera [ID], Marina Casas and Julián Cuevas *[ID]

Department of Agronomy, University of Almeria, ceiA3, 04120 Almeria, Spain;
emilioarredondo.n@gmail.com (E.A.); fmc1984@ual.es (F.M.C.); marinacasas@anecoop.com (M.C.)
* Correspondence: jcuevas@ual.es; Tel.: +34-950-015-559

**Abstract:** Recently there have been new trends in global consumption toward fresh foods that are sources of healthy bioactive compounds, as is the case with pitaya. However, pitaya cultivation is a relatively recent phenomenon and little is known about its management. The objective of this work is to determine the most appropriate annual fruiting pruning method for pitaya in order to obtain a regular annual yield of quality fruit and an intense shoot renewal that guarantee future production. This study compared the response of *Hylocereus undatus* to spur, cane, and combined pruning. As control plants, we left some plants where only sanitary pruning was performed. The results indicate that spur pruning greatly reduced flowering (seven times less than controls) and did not promote intense vegetative growth. Cane pruning, on the contrary, allowed greater flowering which is compatible with a higher number of new shoots (8% more than controls). The vigor of the new shoots was equal in all treatments. Fruit size and quality did not differ either among treatments. Spur pruning only seems applicable as a rejuvenation pruning. Combined pruning gave an intermediate response and seems of no interest given the good shoot renewal provided by cane pruning. Performing sanitary pruning alone may be an interesting option, but only in the first years of cultivation.

**Keywords:** *Hylocereus undatus*; dragon fruit; vegetative growth; flowering; fruit quality





## 1. Introduction

Almost unknown three decades ago, the edible fruit of the cacti of genus *Hylocereus* (Fam. Cactaceae), commonly known as pitaya, pitahaya, or dragon fruit, is currently presented as an interesting and very profitable alternative to traditional crops. Today, it occupies a growing niche in the exotic fruit market, mainly because of its attractiveness, nutritional value, and abundance in bioactive compounds (mainly organic acids and phenols) that provide multiple benefits to human health [1,2]. Pitaya comprises a number of perennial climbing cacti of different species with a complex taxonomy and frequent hybridization between them. Most species of pitaya are characterized by their long green triangular cladodes (stems) which are succulent and articulated (Figure 1A). Their flowers are large, more often white (in some genotypes red) that develop from compound buds (a set of several individual buds), named areoles, formed in the edges of the cladodes (Figure 1B,C). The time elapsed from flower bud formation to anthesis is between 30 and 35 days and between 35 to 42 from bloom to fruit ripening (Figure 1D) [3].

Flowering in *Hylocereus undatus* [(Haw.) Britton and Rose], the most studied species, is seasonal and occurs in waves. In the northern hemisphere, it blooms several times from May to October [4], while in the southern hemisphere it blooms from November to May [5]. Flowering episodes (from four to six depending on the year and location) are cyclical and spread along the season [6]. A period of four to five weeks, depending on the temperature, occurs between flowering waves, during which the fruit develops until maturation [6]. Among the factors affecting pitaya flowering is the age and length of the cladode [7], the

temperature and light, including here radiation intensity and photoperiod [8,9]. In this sense, pitaya is a long-day plant and its floral induction is strongly conditioned by the photoperiod, which regulates a network of biochemical and genetic pathways [10]. Pitaya flowers open for one night only, when, in its native place (Mesoamerica), are pollinated by hawk-moths [11]. A lack of suitable pollinators for pitaya out of its region and self-fertilization barriers due to partial self-incompatibility (not always present) and pronounced herkogamy (Figure 1C) make often mandatory hand pollination (self- or cross depending on the genotype) for the reliable production of commercial fruit.

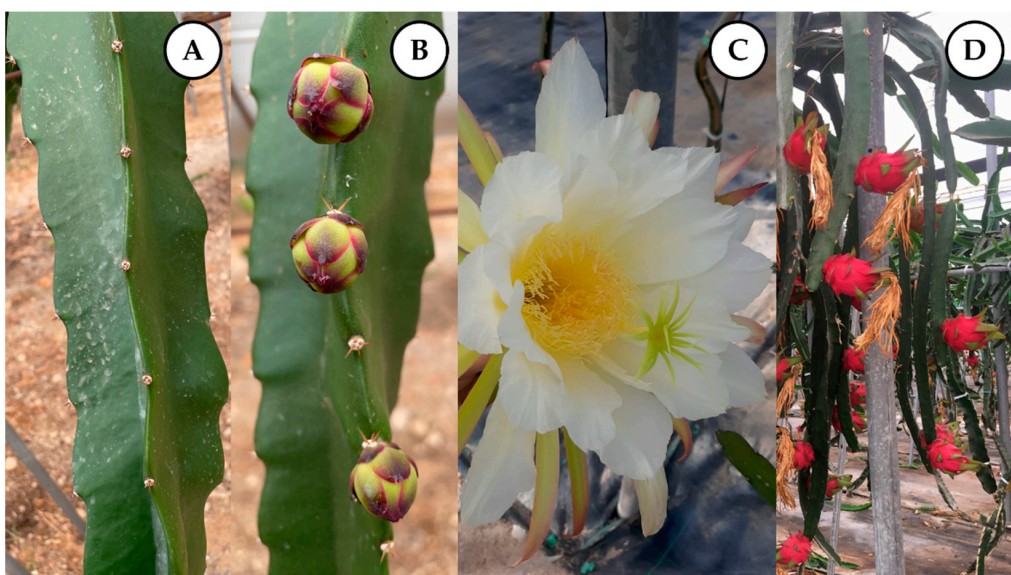

**Figure 1.** Morphological details of pitaya: triangular cladodes (**A**), areoles and floral buds (**B**), flower (**C**), and fruits (**D**).

Despite its recent establishment as a fruit crop of interest in many countries of Asia and America (but also in the Mediterranean area), pitaya management is still to be established in many aspects, including pruning, where recommendations are mostly limited to perform only sanitary pruning [6]. Depending on the age of the plant, three different types of pruning are usually carried out in fruit trees: training, annual (or biennial) fruiting pruning, and rejuvenation pruning in aged plants (when this last is appropriate).

The general objective of annual pruning is to obtain a stable production of fruit of good quality. Pruning, by removing wood, implies inevitably cutting down some flowers buds and thus sacrificing to some extent the maximum production of fruit to ensure its optimum quality and a more constant yearly production [12]. In pitaya, as in other fruit crops, annual pruning requires cleaning operations of pest damaged and broken wood and limiting the size of the plant to the dimensions established according to the planting frame, in order to avoid the negative effects of excessive shading on flowering. In addition, it is necessary to adjust fruit load and to renew the fruitful shoots. However, the way to proceed to adjust fruit load and to renew shoots is not determined in pitaya, since the scientific information published on pitaya pruning is so far very scarce [13]. Taking into consideration pruning carried out in other climbing crops such as table grape and kiwi vines, we decided to compare flowering, yield and fruit quality under three different pruning methods: spur, cane, and combined (i.e., a mixture of spur and cane pruning).

The objective of this work is to determine the most appropriate annual fruiting pruning method for pitaya cultivation in order to obtain an optimal annual production, with a timely production of flowers, fruits, and a sufficient shoot renewal that guarantee future production.

## 2. Materials and Methods

### 2.1. Experimental Site and Crop Management

This study was performed in a pitaya [*Hylocereus undatus* (Haw.) Britton and Rose] orchard of cultivar 'Korean White' located at the UAL–Anecoop Foundation, in Almeria (SE Spain) (longitude 2°17′02″ W, latitude 36°51′54″ N). The altitude is 95 m above sea level and the orchard is 5 km away from the Mediterranean Sea. The experimental area presents a semiarid subtropical Mediterranean climate according to the agroclimatic classification of Papadakis [14], with an average annual temperature of 18.5 °C.

December and January are the coolest months in the area, while August is the warmest. Rain averages 250 mm per year (January to December), while the mean annual relative humidity oscillates between 67% and 73% depending on the year. Bright sunny days are common at the experimental site. Sunlight hours reach a mean value of 3273 h per year. The photoperiod that determines flower induction in this species oscillates in our experimental site between 14 h/10 h (day/night) in summer and 10 h/14 h in winter. Since a critical photoperiod of 12 h has been established for red pitaya [8], inductive day-length in our latitudes starts at mid-March and finishes at mid-October. The soil in the experimental plot is a sandy clay loam with 46.4% sand, 20.6% silt, and 33.0% clay, according to sampling carried out at 20 cm depth, since most roots in pitaya are found in the first 40 cm [15].

The pitayas used for this study were four-year-old plants of the cultivar 'Korean White' (also known as 'K1'). 'Korean White' is a self-compatible cultivar, but, however, still requires hand pollination given the absence of natural pollinators in the experimental site and the marked herkogamy these pitayas exhibit (Figure 1C). Hand pollination was performed early in the morning (from 5:30 to 7:00 a.m.) at flower anthesis collecting first fresh pollen of open flowers of nearby plants of the same cultivar and applying it using a fine paint brush on the stigma of each open flower. The experimental pitayas were planted from cuttings (no rootstock included) in 2017 and then they were in full production. The plants were annually fertirrigated with doses of 146 UF of nitrogen, 100 UF of phosphorous and 470 of potassium applied proportionally to the volume of irrigation water applied of circa 3000 m³ ha⁻¹, distributed by drip irrigation as monthly represented in Figure 2. Pest and disease control was carried out in the orchard following IPM guidelines. All cultural practices except for pruning were performed the same way on all of the experimental plants.

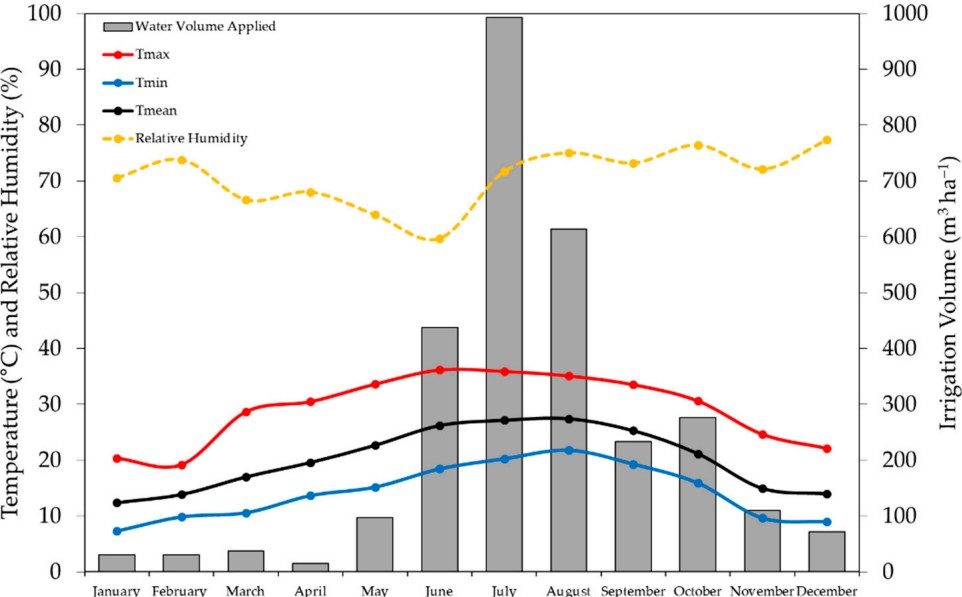

**Figure 2.** Monthly average temperatures, relative humidity, and irrigation volumes applied at the experimental site.

The pitayas were located under a greenhouse structure and arranged at a spacing of 1 m between plants and oriented following a two-dimension trellis system (similar to a "Geneva" double curtain) 1.8 m height (Figure 3). From this height, the cladodes hung forming a production wall. The greenhouse was a flat roof structure covered with polyethylene plastic film of 0.2 mm, placed 2.7 m above the top of the canopy of the experimental plants.

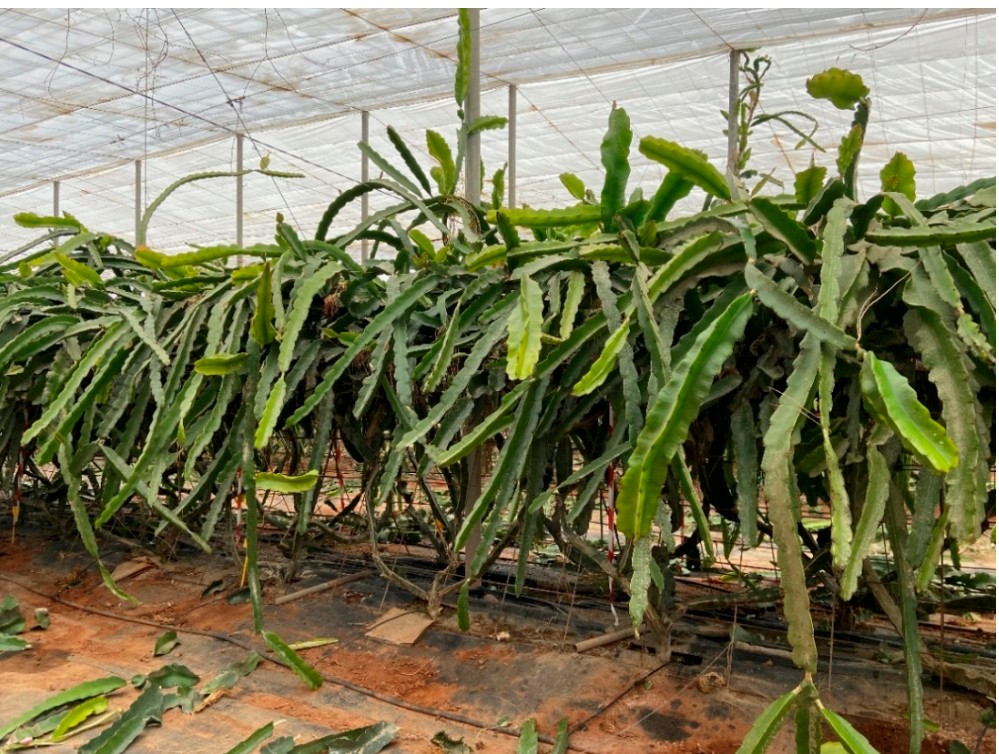

**Figure 3.** Partial view of the greenhouse and pitaya training system.

### 2.2. Treatments and Experimental Design

The study compared the response of *H. undatus* to three different methods of pruning: spur, cane and combined. Spur pruning consisted of eliminating approximately two thirds of each cladode, leaving only the four to five basal areoles or compound buds (in each cladode edge). Cane pruning consisted of just tipping all cladodes, cutting back only the last apical buds. In the combined pruning, a mixture of both types of pruning was used, with half of the cladodes spur-pruned and the other half cladodes cane-pruned. Pruning was carried out on 26 April 2021. As with the control plants, we left some wall sections where only sanitary pruning was performed in March, leaving the productive cladodes intact and removing only decayed cladodes.

The experimental design was completely randomized with four replicates per treatment. To randomize replicates, the 50 m length Geneva double curtain (25 m in the north side, and 25 m in the south one) was divided into sixteen 2.5 m length segments, leaving four more at both ends of the curtain to avoid edge effect. The sixteen segments were randomly assigned to the treatments (four per each) and acted as replicates. After pruning, a sample of 12 cladodes per replicate were tagged and followed.

### 2.3. Measured Parameters: Vegetative Growth, Flowering and Fruit Quality

On tagged cladodes, we assessed their vegetative and reproductive performance. In this sense, we recorded and compared the number and length of the new growing shoots formed, the intensity of flowering (flowers per cladode) and the number of fruit and its size and quality [(weight, equatorial diameter, length and total soluble solids (TSS)]. Yield

per cladode in each harvest episode was also calculated taking into account the number and average weight of the fruit.

The number and length of the new vegetative shoots were determined, regardless their length, at 50, 100, and 158 days after pruning (DAP), i.e., 15 June, 4 August and 1 October, respectively. New growing cladodes were counted at each date and its vigor estimated by its average length, measured with a seamstress meter. Flowering intensity, as the number of flowers (and developing buds flower near anthesis) per cladode, was determined at 80, 110, 136, and 165 DAP (15 July, 14 August, 9 September and 8 October, respectively), corresponding to the flowering waves of the experimental plants. Fruit set as the percentage of fruit obtained after hand cross-pollinating flowers was calculated for each flowering wave separately. Given the non-climacteric nature of pitaya, harvesting was carried out at fruit ripening, assessed by the full development of magenta color in the fruit skin (Figure 1D). Size and TSS content were measured on all harvested fruit. The size of each single fruit at harvest was estimated by its weight, length and equatorial diameter. TSS content was measured using a digital refractometer (model PAL-1, Atago Co., Tokyo, Japan) from the juice of a central portion of each fruit; data were expressed in °Brix. Parameters were compared by analysis of variance (ANOVA). When necessary, means were separated by Tukey's test at 5% ($p < 0.05$) using AgroEstat software, version 1.1, São Paulo State University (UNESP), Jaboticabal, Brazil [16].

## 3. Results

### 3.1. Vegetative Growth

Pruning had a depressive effect on new growth. This depressive effect led to fewer new cladodes of less vigor. The effect was more noticeable the more intense the pruning (Table 1). Spur pruning, by reducing the number of buds left after cutting back the cladodes, significantly reduced the number of new shoots at the end of the season, with a reduction of more than 50% with respect to cane pruning and control plants (Table 1). Cane pruning slightly reduced the number of new shoots with respect to control plants, although there were no statistical differences. Combined pruning achieved an intermediate situation (Table 1). Most new growth took place in the first 100 DAP, with less growth late in the season, when temperature and radiation declined in our experimental site. As the season progressed, the distinction between new shoots and one-year old cladodes and older was more difficult.

**Table 1.** Number of new cladodes and length (cm) along the season in response to different pruning treatments.

| Treatments | Number of New Shoots | | | New Shoots Length | | |
|---|---|---|---|---|---|---|
| | Days after Pruning (DAP) | | | | | |
| | **50** | **100** | **158** | **50** | **100** | **158** |
| Control | 2.0 a | 2.3 a | 2.4 a | 30.9 a | 42.9 a | 47.0 a |
| Cane | 1.8 a | 2.2 a | 2.3 a | 27.1 a | 53.1 a | 52.0 a |
| Combined | 1.1 a | 1.4 ab | 1.5 ab | 32.2 a | 52.4 a | 69.2 a |
| Spur | 1.0 a | 1.1 b | 1.1 b | 21.8 a | 48.5 a | 72.2 a |
| *p* | 0.0854 | 0.0281 | 0.0250 | 0.1391 | 0.2951 | 0.2863 |

50 DAP = 15 June; 100 DAP = 4 August; 158 DAP = 1 October. Mean comparison in columns by Tukey's test at $p < 0.05$. For each column, different letters indicate a statistical significance.

Given the intensity of spur pruning, we expected that the vigor of the new shoots (fruiting organs in the following year) estimated by their length would be greater than those formed in control and cane pruned plants. However, no statistical differences in the vigor of the new shoots were observed in any date (Table 1). In this sense, the vigor of the new cladodes was slightly, but not significantly, increased by more severe (spur) pruning (Table 1). This was, in part, due to the variability observed in the length of the new cladodes, especially in cane-pruned plants. A different situation occurred when we calculated new

growth in total. That is, the total length of all of the new cladodes, and therefore the fruiting potential for the next season. In this case, there were statistical differences favoring control and cane-pruned plants against spur-pruned vines in the second measurement date (Figure 4). However, at the last assessment (158 DAP), the differences were attenuated and there were no longer significant differences in total shoot length among treatments. As expected, combined pruning reached again an intermediate result (Figure 4). It should be noted that these measurements took into account all of the shoots emitted as new growth, and shoots of very different length were observed, ranging from very short, recently formed cladodes, to very long cladodes with vigorous growth. This led to high variability within the treatments (as explained above).

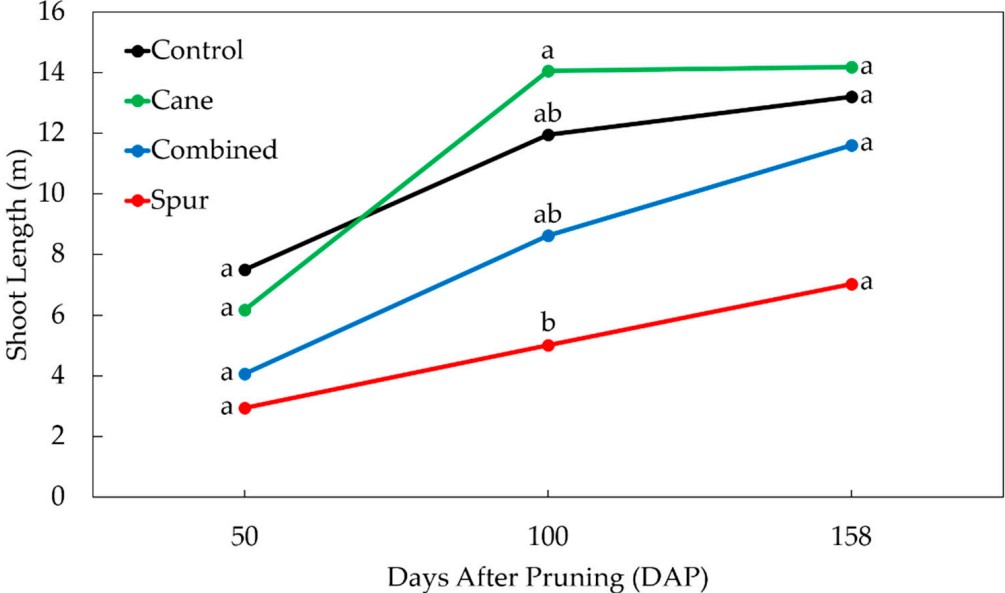

**Figure 4.** Changes along the season in total new growth as a result of the number and length of new shoots in response to different pruning treatments. Mean comparison for dates after pruning carried out by Tukey's test at *p* < 0.05. For each date, different letters indicate statistical significance.

### 3.2. Flowering

The first wave of flowering took place at 80 DAP. A higher flowering intensity in this first wave was observed in caned-pruned and in control plants, and a lower flowering intensity was observed in combined and spur pruning treatments (null in the latter) (Table 2). As before with vegetative growth, the intensity of pruning reduced flowering to a greater extent the more severe the intervention was (to the point where spur pruning completely eliminated the appearance of flowers in this first wave) (Table 2). Although the differences in flowering between cane pruning and control were not significant, it is interesting to note that tipping cladodes (cane pruning) seemed to promote a little more flowering with respect to control plants (Table 2). Cane pruning also showed a positive effect on bloom earliness, since the first day of observation in this wave (80 DAP) 50.0% of the flowers were open (the rest were advanced flower buds), while control plants had 35.7% open flowers (the remaining still developing closed buds) and the combined pruning treatment had only 14.3% of flowers in anthesis; the rest were flower buds yet to open. The spur pruning treatment had no flowers in this first wave (Table 2).

The second wave of flowering occurred at 110 DAP, 30 days after the first wave. In this second wave, the highest number of flowers per cladode was again obtained in cane pruning, treatment in which tipping favored again a higher intensity of flowering (Table 2). In this bloom episode, spur pruning treatment produced some flowers, although its flowering level was again significantly lower than that of cane pruning (Table 2). In combined pruning and in control plants, the same flowering intensity was observed, with combined pruning showing a strong increase in this second flowering with values not

statistically different from those of cane pruning (Table 2). In the third and in the fourth wave of flowering at 136 and 165 DAP, there were no differences between treatments, with a strong reduction of the intensity of flowering compared to previous waves. Flowering also became more desynchronized and scattered. In the fourth wave, flowering was very low and erratic in all treatments (Table 2).

**Table 2.** Bloom intensity (flowers/cladode) along the season in response to different pruning treatments.

| Treatments | Days after Pruning (DAP) | | | | |
|---|---|---|---|---|---|
| | 80 | 110 | 136 | 165 | Total |
| Control | 0.5 a | 0.5 ab | 0.4 a | 0.1 a | 18.8 ab |
| Cane | 0.7 a | 0.6 a | 0.3 a | 0.1 a | 20.3 a |
| Combined | 0.2 ab | 0.5 ab | 0.1 a | 0.1 a | 11.0 b |
| Spur | 0.0 b | 0.1 b | 0.1 a | 0.0 a | 2.5 c |
| *p* | 0.0049 | 0.0447 | 0.1480 | 0.6158 | <0.0001 |

80 DAP = 15 July; 110 DAP = 14 August; 136 DAP = 9 September; 165 DAP = 8 October. Mean comparison in columns by Tukey's test at *p* < 0.05. For each column, different letters indicate a statistical significance.

The effects of pruning levels on flowering intensity were clearly expressed at the total level of flowering at the end of the year, with cane pruning treatment producing 8.0% more flowers than control plants, 84.6% more flowers than combined pruning, and eight times more flowers than spur pruning treatment (Table 2).

### 3.3. Fruit Quality and Yield

The first crop reached full ripening at 14 August, 30 days after flowering (DAF) and 110 DAP, when fruit size and quality were measured (Table 3). It should be noted that the number of fruits harvested was practically coincident with the number of flowers formed in each pruning treatment, indicating that under our experimental conditions fruit set after hand pollination was close to 100%. Therefore, more fruits were harvested in cane pruning treatment, underlining the effectiveness of terminal bud tipping that favored the development of floral buds and the formation of abundant fruits. On the contrary, the severity with which spur pruning was carried out resulted in no fruit at this date, since pruning eliminated most of the flower buds. In the combined pruning treatment, some fruits were harvested, especially in those cladodes that were cane-pruned (Table 3).

**Table 3.** Fruit production per cladode and fruit characteristics as an average in the first harvest in response to different pruning treatments.

| Treatments | Fruits/Cladode | Weight (g) | Equatorial Diameter (mm) | Length (mm) | TSS (°Brix) |
|---|---|---|---|---|---|
| Control | 0.5 a | 472.5 a | 85.3 a | 120.9 a | 13.6 a |
| Cane | 0.7 a | 434.7 a | 85.6 a | 115.1 a | 13.7 a |
| Combined | 0.2 ab | 498.5 a | 87.8 a | 119.1 a | 13.9 a |
| Spur | 0.0 b | – | – | – | – |
| *p* | 0.0071 | 0.7283 | 0.9290 | 0.7292 | 0.5535 |

Mean comparison in columns by Tukey's test at *p* < 0.05. For each column, different letters indicate a statistical significance.

Crop quality did not differ among treatments. Thus, fruit size and sweetness were similar in all treatments. This is to say that the lower yield in some pruning treatments did not result in improved quality or size in any of the harvests evaluated (Tables 3–5). In this respect, differences in fruit size were small, with only a slight increase in weight and equatorial diameter in the fruit of plants subjected to combined pruning. The same trend was observed in fruit sweetness, a parameter in which a small improvement was observed in combined pruning compared to the fruit obtained in cane pruning and control treatments (13.6 vs. 13.9 °Brix; Table 3).

Fruits from the second wave of flowering reached full maturity at 33 DAF (143 DAP) and showed the same trends noted above. Harvesting of those first fruits from the plants

that were spur-pruned took place then (Table 4). A lower fruit set was observed now in cane and combined pruning (49 and 43%) than in control (76%) and spur-pruned cladodes where all flowers set fruit. Yield per cladode did not differ, in part since the enhanced fruit set in spur pruning treatment and partly as a result of the large variability observed among the replicates of the different treatments (Table 4). Fruit quality was again unaffected by pruning intensity, although there was a significant and generalized reduction in fruit size (weight, equatorial diameter and length) and sweetness (°Brix) was observed (Table 4) in this second harvest compared to the fruit from the first harvest (Table 3).

**Table 4.** Fruit production per cladode and fruit characteristics as an average in the second harvest in response to different pruning treatments.

| Treatments | Fruits/Cladode | Weight (g) | Equatorial Diameter (mm) | Length (mm) | TSS (°Brix) |
|---|---|---|---|---|---|
| Control | 0.4 a | 408.1 a | 76.2 a | 125.1 a | 10.3 a |
| Cane | 0.3 a | 369.4 a | 73.6 a | 111.1 a | 11.2 a |
| Combined | 0.2 a | 417.1 a | 77.0 a | 106.5 a | 11.3 a |
| Spur | 0.2 a | 384.5 a | 77.8 a | 113.3 a | 12.4 a |
| *p* | 0.2382 | 0.8261 | 0.7828 | 0.4484 | 0.4184 |

The second harvest of fruit reached full maturity on approximately 16 September, 33 DAF and 143 DAP. No significant differences occurred in any parameter after mean comparison by Tukey's test at $p < 0.05$.

As temperatures dropped in autumn (Figure 2), reaching fruit maturity took a longer period. On these dates, flowering was also scattered and there was some overlap between the third and fourth waves of flowering and their harvests. Late fruit from the third wave of flowering reached maturity between 42 and 53 DAF (between 178 and 191 DAP) and confirmed that fruit quality was not affected by pruning treatments (data not shown). Finally, a last weak episode of flowering took place at 8 October that confirmed the downward trend in fruit size for the control and also for spur and cane pruning treatments, while an improvement was observed in combined pruning treatment (Table 5). Total soluble solids content (°Brix) was the only parameter that showed a clear increase in all treatments, exceeding the values measured in the first harvest (Table 5).

**Table 5.** Fruit production per cladode and fruit characteristics as an average in the last harvest in response to different pruning treatments.

| Treatments | Fruits/Cladode | Weight (g) | Equatorial Diameter (mm) | Length (mm) | TSS (°Brix) |
|---|---|---|---|---|---|
| Control | 0.2 a | 374.5 a | 75.4 a | 102.7 a | 16.2 a |
| Cane | 0.2 a | 326.0 a | 75.9 a | 95.0 a | 14.3 a |
| Combined | 0.3 a | 515.2 a | 87.6 a | 114.3 a | 14.9 a |
| Spur | 0.2 a | 400.4 a | 75.1 a | 102.6 a | 15.0 a |
| *p* | 0.2704 | 0.4526 | 0.4197 | 0.5902 | 0.6528 |

The last harvest of fruit reached full maturity on approximately 3 December, 56 DAF and 221 DAP. No significant differences occurred in any parameter after mean comparison by Tukey's test at $p < 0.05$.

Finally, the higher abundance of fruit of the same weight in cane-pruned and control plants in the first and second harvests lead to a heavier total yield per cladode in these two treatments, with important differences with respect to spur-pruned plants at the end of the season. In this regard, an average of 463 and 456 g of fruit per cladode were harvested after pruning in control and cane-pruned plants (13 fruit each harvested in 12 cladodes) versus 158 g per cladode in spur-pruned plants (6 fruits in all 12 cladodes). Combined pruning showed again intermediate results (327 g per cladode; 9 fruits in total in all 12 tagged cladodes) with a noticeable increase in yield in the last harvests.

## 4. Discussion

Gunasena et al. [13] indicate that a well grown pitaya plant in the first year may produce about 30 branches (cladodes), increasing to 130 in the fourth year. This strong growth causes intense shading that reduces light penetration into the canopy and makes more difficult cultural practices (especially hand pollination, pest control, and harvesting). On the contrary, Crane and Balerdi [17] note that, under conditions in Florida, pruning after last harvest favors next year bloom and bud growth, so annual pruning would be advisable in order to promote new shoot growth and maintaining healthy plants with well illuminated and vigorous stems.

Sánchez [18] recommend annual pruning in pitaya limited to a sanitary and canopy size adjustment, eliminating only those cladodes growing up vertically and favoring shoots that grow horizontal on the trellis and those hanging down that result more accessible to hand pollination. The aim is to keep the hanging cladodes well aerated and illuminated, pruning those that grow inwards, out of the operator's reach or showing severe damages. However, this recommendation does not indicate how to operate at the level of the cladode and does not consider the necessary renewal of fruiting organs, the cladodes that guarantee future production. The method of perform pruning and any load adjustment that may be necessary are not then indicated.

Given the condition of climbing cactus and the lack of specific information about the best pruning system for pitaya, here we have compared spur, cane, and combined pruning usually performed in other vines such kiwi and table grape. In table grapes, the main objective of fruit pruning is to achieve a fairly constant yield of high quality that must remain constant over time. This requires ensuring a balance between vegetative growth, bearing future crop, and current fruit load. In addition, pruning has to maintain the size and shape of the vine decided when training young plants and according to the assigned planting frame [12]. This balance is sufficiently achieved in pitaya by cane pruning due to the fact that, while maintaining a higher number of flower buds after pruning (Table 2), new shoot growth is more than enough in cane-pruned plants (Figure 4). In kiwi, winter annual pruning of female plants requires leaving long canes since female flowers also develop laterally, and fruit load would be limited if the canes are shortened too much by pruning. The shoot renewal in female kiwis is achieved by selecting as production wood those canes arising close to two-year-old branches. However, one-year-old spurs are also left after pruning since they produce a sizable yield [19]. Marodin et al. [20] compared growth, yield, and kiwi fruit quality after pruning the productive one-year-old shoot to different length and recommend leaving 10-bud canes. Canes left with 20 buds produce more fruit but of lower size. This negative effect on fruit quality has not been observed in our experiments, where fruit size and sweetness were the same in all treatments. On the contrary, in table grape a too heavy fruit load obtained with long pruning might lead to poor fruit size and delayed ripening. Thus, an adjustment of the number of fertile buds left after pruning is needed in grape. Depending on the genotype, long cane pruning may result too in poor shoot renewal lowering yield next season. On the contrary, combined pruning allows usually sufficient and stable yield and at the same time good shoot renewal [21]. A fundamental difference between pitaya and table grape and kiwi vines is that these latter are deciduous, while pitaya maintains reserves in the evergreen cladodes.

Our results clearly show that long pruning in pitaya provides more flowers and more fruits of the same quality than short pruning. The combined pruning provided intermediate results. This can be explained due to the fact that cane pruning preserves more flower buds in the areolas where the pitaya blooms. Pitaya exhibits a strong apical dominance with only occasional branching. In fact, cane pruning stimulated sprouting by removing the terminal buds and thus slightly improved flowering levels compared to control plants, in which cladodes were not tipped.

Winter pruning after last harvest is recommended by Le Bellec et al. [6] to favor the growth of new cladodes and the multiplication of the areoles where the flowers are formed in the following season. However, our study did not show an increase in the number

of shoots when heavy pruning (i.e., spur pruning) was carried out. On the contrary, it was found that cane pruning, aimed at higher fruit production, also promoted greater vegetative growth compared to the other treatments, at least initially, with a higher number of new shoots formed. The reasons for this result probably have to do with the fact that pruning removes cladodes, that are the photosynthetic organ (the source) in pitaya, so more flowers and fruits (the sinks) are compatible with cane pruning. Besides, in pitaya, the cladodes accumulate most reserves that do not migrate to older wood and root, as occurs in deciduous fruit trees.

Although there were differences in the number of shoots in the different treatments, favoring cane pruning and control plants, all showed equivalent vigor. In deciduous fruit trees is usual that heavy pruning provides greater vigor of the new shoots formed [12], but in our case, by eliminating photosynthetic active cladodes, the differences in vigor were not initially favorable to spur pruning (Table 1 and Figure 4). Nonetheless, in the second and third measurements, an increase in the vigor of new shoots was observed in spur-pruned cladodes, reaching and even surpassing at the end the vigor observed in the other pruning treatments (Table 1). On the contrary, by eliminating a significant number of buds, spur pruning lost the first flowering and there was a strong reduction of the second flowering wave. Combined pruning reached an intermediate level of flowering, since it reduced flowering intensity in each flowering episode, although the differences did not reach statistical significance until the final estimation of bloom intensity was calculated (Table 2).

The first harvesting mirrored the first wave of flowering, leading control and cane pruning to yield more fruits. The near 100% fruit set suggests an easy setting where hand pollination is performed in these flowering waves but is less effective in the later bloom episodes. In spite of the self-compatibility of 'Korean White', late flowers were less fertile despite the fact that the hand pollination procedure followed was the same. This suggests some physiological bases, besides pruning, in the lower fruit set obtained at the end of the season. With reference to fruit weight, it can be concluded that they were very similar, with an average fruit weight close to 500 g, within the range reported by Vaillant et al. [22] and Ochoa-Velasco et al. [23], which values vary between 200 and 570 g depending on the genotype. The first harvest was the most important in term of yield. From the second harvest, a downward trend began and the last harvest was not relevant in terms of quantity. Other fruit size parameters (fruit length and diameter) were also similar in all treatments suggesting that fruit size in pitaya depends more of successful pollination (number of seeds formed) than on canopy size or sink/source relationships. In other words, lower yield did not result in larger, heavier and/or sweeter pitayas. Warusavitharana et al. [24] indicate that the average fruit length of *H. undatus* is 113 to 142 mm as here was found. As for TSS content, average values of 13 °Brix have been reached, within the range between 12 and 14 °Brix reported by Mercado-Silva [25], with some fruit reaching 15 °Brix, as reported by Chien and Chang [26] and Goenaga et al. [27]. Since the cladodes in cane pruning and control plants produced more fruit of similar weight, yield was largely increase in these two treatments, while a poor harvest was produced in spur-pruned plants.

## 5. Conclusions

Cane pruning is deemed as the best pruning option under the conditions of this trial (i.e., plants in full production), since it allows a high intensity flowering, leading to a higher fruit production. It was also compatible with a greater number of vegetative shoots, of the same vigor that in the other pruning treatments.

Spur pruning, due to its severity, strongly reduces first and second flowering and, at the same time, does not promote more intense vegetative growth that can be considered as an advantage for the renewal of fruiting organs. Its only justification could be when a rejuvenation process of old pitaya vines is intended after several years of cultivation.

Combined pruning does not seem to provide any benefit over cane pruning, since it leads to an intermediate response in most parameters.

Finally, the control plants produce similar results to those of the cane pruning ones. Therefore, performing a light sanitary pruning seem to be feasible for some years and represents a more economical option, at least for the first years of cultivation, by eliminating the cost of more detailed pruning.

Once the most productive methods for perform annual pruning have been established, the following research goals for us would be the estimation of the number of productive shoots per area unit that should be left after pruning and how this level of pruning could affect both current and future yields.

**Author Contributions:** Conceptualization, J.C.; methodology, J.C.; validation, M.C.; formal analysis, E.A. and F.M.C.; investigation, E.A., F.M.C., M.C. and J.C.; resources, J.C.; data curation, E.A., F.M.C. and J.C.; writing—original draft preparation, E.A.; writing—review and editing, F.M.C. and J.C.; supervision, J.C. All authors have read and agreed to the published version of the manuscript.

**Funding:** This research received no external funding.

**Institutional Review Board Statement:** Not applicable.

**Informed Consent Statement:** Not applicable.

**Data Availability Statement:** All tables and figures are original and all data is given in the article.

**Acknowledgments:** We would like to thank the Anecoop (Almeria, Spain), which kindly provided means and personnel.

**Conflicts of Interest:** The authors declare no conflict of interest.

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
