# Peer review of "Comparing Different Methods for Pruning Pitaya (Hylocereus undatus)"

_horticulturae, doi:10.3390/horticulturae8070661_

Round 1
Reviewer 1 Report
The work analyses the effect of different types of pruning in the species Hylocereus undatus. The study has a linear approach but can provide some indications on the management of the plant in cultivation.
The work is well structured, but it has some flaws that require major revision.
The title is too concise. I would suggest thinking something like 'Influence of pruning on productive behaviour of pitahaya (Hylocereus undatus, Haw.)'. Please, check the correct name, pitaya or pitahaya. Let consider also to use 'dragon fruit' as vulgar name.
Since the species is not well known, it would be useful to include a paragraph in the introduction briefly describing the cultivation method.
Lines 110-112: please, check the comment on the pdf
Lines 112-118: the description of the traits seems to be referred to the measured parameters described in the paragraph 2.3. If so, please move it in the paragraph
An important issue concerns the fruit set. The authors should specify (in M&M) if the species is self-incompatible and if the pollination was natural, or manual for all the harvest. Please, on these issues check the comment at the lines 208-209, and 228-232.
English needs a revision.
Further comments can be found in the attached pdf.

Author Response
Answers to Reviewer 1 in bold after his/her comments
The work analyses the effect of different types of pruning in the species Hylocereus undatus. The study has a linear approach but can provide some indications on the management of the plant in cultivation.
Thanks for your kind comments
The work is well structured, but it has some flaws that require major revision.
The title is too concise. I would suggest thinking something like 'Influence of pruning on productive behaviour of pitahaya (Hylocereus undatus, Haw.)'.
We modified the title to make it more comprehensive
Please, check the correct name, pitaya or pitahaya. Let consider also to use 'dragon fruit' as vulgar name.
We added them as synonymous, so readers have no confusion about the species we are working with. The new title makes it clear too.
Since the species is not well known, it would be useful to include a paragraph in the introduction briefly describing the cultivation method.
We developed more cultivation methods of the experimental plants in M&M section. In Introduction, we added some basic information about flowering that complement tha e information included in M&M.
Lines 110-112: please, check the comment on the pdf
Correct. You are right. We have 50 m length in total, 25 m I each of the two sides of the double curtain. We leave this clear know modifying the sentence
The 50 m length Geneva double curtain (25 m in the north side and 25 m in the south one) was divided into twenty 2.5 m length segments. Sixteen segments (four per treatments) were used as replicates for the experiment and we left segments too and at the start and end of the curtain (4) to avoid edge effect.
Lines 112-118: the description of the traits seems to be referred to the measured parameters described in the paragraph 2.3. If so, please move it in the paragraph
You are right, the sentence described parameters. I move this sentence at the beginning of the next subsection.
An important issue concerns the fruit set. The authors should specify (in M&M) if the species is self-incompatible and if the pollination was natural, or manual for all the harvest.
This genotype (Korean White) is self-compatible. However, the so called Queen of the Night flower exhibits a marked herkogamy, with the stigma well above the anthers, and since the pollinating agents are not present in Spain, we rely on hand pollination performed from 5:30 to 7 am in the morning. We carried out hand pollination using fresh pollen of flowers opening at the same night by means of a fine paint brush. I added this information in the manuscript in M&M. However, 100% fruit set is not always guarantee since female fertility might change along the season as actually does, because we performed hand pollination similarly and the results are different. This happens every year in the management of the crop. First flowering waves are more fertile.
Please, on these issues check the comment at the lines 208-209, and 228-232.
Attending your suggestions, an explanation was added in the M&M section about hand pollination procedures. Fruit set was simply calculated as the percentage of flowers setting fruit and we explain this now in M&M.
English needs a revision
Further comments can be found in the attached pdf.
Thanks. We carefully checked language and comments. I have taken into considerations most comments included in the pdf file, but those regarding flower formation in spur pruned plants where you refer to the plants organs acting as source. In this regard, in this species, the source for photoassimilates is, as we both know, the cladodes, that when severely pruned not only eliminated potential flower buds, but also limited the source (different in deciduous fruit crops). Of course, next year if severely pruned again (as performed in grape) the same results are obtained. In grape, spur pruning is possible because fruit clusters come in new growth from basal buds, not in “old” (1-year-old) wood.
I have tried to clarify all other comments added in the pdf file that I found very useful for submitting a better version. I hope you agree. Thank you very much again for your useful suggestions.
Reviewer 2 Report
The authors performed the experiment which not bring any significant differences in the results. Appropirate methodological assumptions and correct measurements of plant growth and yield were made. A more detailed comparision of the obtained results should be made with the results obtained by other authors regarding pruning for example the species proposed by the authors themselves. Othervise, everything is based on the deliberations of the authors, albeit correct, however scientific publication require a greater depth in to the subject, in this case pruning of plants.

Author Response
Answers to Reviewer 2 in bold after his/her comments
The authors performed the experiment which not bring any significant differences in the results. Appropirate methodological assumptions and correct measurements of plant growth and yield were made. A more detailed comparision of the obtained results should be made with the results obtained by other authors regarding pruning for example the species proposed by the authors themselves. Othervise, everything is based on the deliberations of the authors, albeit correct, however scientific publication require a greater depth in to the subject, in this case pruning of plants.
Thank you I took into consideration your suggestions and developed more and compared pruning results with that of table grape and kiwis as suggested.
The title was modified
We have also completed and edited the manuscript where needed as indicated in the pdf file accompanying your suggestions.
Finally, we added too in Conclusions some indications about where the research should focus in the next future as suggested too.
Round 2
Reviewer 1 Report
The revisions have improved the work, but there are still some aspects to be discussed.
English still needs to be improved.
Further comments are in the attached pdf.

Author Response
Answers to Reviewer
Line 62. Evaluate also “formative”
Training is more commonly used.
Line 139. The sixteen segments were randomly (?) assigned to the treatments (four per each) and acted as replications
Accepted suggestions.
So, if all harvested fruits were measured, the sample size wold vary?
Well, the subsample size varied, but we used the replicate average and we have the same number of replicates per treatment.
Are there any literature references or an internet link?
Different situation occurred when we calculated new growth in total; that is, the total length of al the new cladodes
Modified as proposed
The period remains unclear
80DAP. The date is also included in Table 2.
Please, solve the comment I made in previous version
The table tittled “Fruit production per cladode and fruit characteristics…” includes fruit production and fruit size parameters (weight, length and diameter) and soluble solid content. I do not find any better word than fruit characteristics to define using a single word fruit size and sweetness. It’s not a big deal however.
The variety is self-compatible and the flowers are hand-pollinated, but all the treatments had the fruit-set lowered. Therefore, baside the effect of pruning it can be present a physiological effect
Yes. As stated late flowers seems to be less fertile, and that occurred in all treatments. We added the following sentence in the discussion to make it clear.
“Despite self-compatibility of Korean White, late flowers were less fertile despite hand pollination procedure followed was the same. This suggests some physiological bases, beside pruning, in the lower fruit set obtained at the end of the season.”
To give the reader an idea of “higher abundance, perhaps a column with the number of fruits shoud be added.
We have considered this, but we opted for simplicity to add yield per cladode in the text. However, attending reviewer petition, we have now included also the average number of fruit per cladode.
I am very very sorry, but I don’t see why is difficult to understand. Nonetheless, I have modified the sentence to make the differences ever more clear.
Branches?
No. Canes. Branch (or branches) in Pomology is more often use for Wood older tan 2 years. Cane is here a kind of 1-year-old shoot. Cane is the correct term.
Please, see the previous comment
Please, see my answer above
the first harvesting mirrored the first wave of flowering, leading control and cane pruning to yield more fruits
Accepted and included
Quantity. Modified accordingly
I don’t see the contradiction. Under correct management, fruit size in pitaya depends of sink strength determines by the number of seed included due to good fertilization. We have preliminary results no yet published that probe a linear very significant relationship between fruit weight and number of seed at harvest. I can add between parenthesis Cuevas et al., in prep., but I don’t feel is appropriate and not a matter of this experimentation.
Yes. Better word. Thanks.
the following research goals would be the estimation of the number of productive shoots per area unit should be left after pruning and how this level of pruning could affect current and future yields
Ok. Included as suggested.

Reviewer 2 Report
The authors responded to most of the comments. Publications in this form is suitable for further processing.
Author Response
Thank you for your help